# Charity-Provided Community-Based PSA Testing for Assessment of Prostate Cancer Risk in the UK: Clinical Implications and Future Opportunities

**DOI:** 10.3390/cancers17101728

**Published:** 2025-05-21

**Authors:** Artitaya Lophatananon, Graham Fulford, Jon Young, Susan Hart, Matthew Brine, Kenneth R. Muir

**Affiliations:** 1Division of Population Health, Health Services Research and Primary Care School of Health Sciences, Faculty of Biology, Medicine and Health, The University of Manchester, Manchester M13 9PL, UK; artitaya.lophatananon@manchester.ac.uk; 2GFCT Ltd. Formerly, The Graham Fulford Charitable Trust, 66b Smith Street, Warwick CV34 4HU, UK; 3Empresa Limited, 43 All Saints Green, Norwich NR1 3LY, UK

**Keywords:** PSA community testing, risk stratification, riskman algorithm, early detection

## Abstract

Prostate cancer is the most common cancer in UK men, yet the lack of a national screening program results in inconsistent early detection. The Graham Fulford Charitable Trust (GFCT) delivers community-based PSA testing to improve access and awareness. This study analysed GFCT data from 2021 to 2024 to evaluate how well its traffic light risk system identifies men at increased risk. The model uses age-specific PSA thresholds and groups individuals into green, amber, or red risk categories. Men in the red group had over 15-fold higher odds of clinically significant cancer (Grade Group ≥ 3) compared to those in the green group. Based on self-reported data collected in this survey, the GFCT approach performed comparably to other approaches, such as the multivariable “Riskman” score, which had a slightly higher AUC (0.84 vs. 0.76). Riskman benefited from integrating the PSA free-to-total ratio. The GFCT now performs over 50,000 tests annually and is evolving to include reflex biomarkers and GP follow-up options, supporting shared decision-making. Whilst not a substitute for formal screening, the GFCT model complements research efforts like the UK TRANSFORM trial and highlights the potential of low-cost, community-based approaches to improve early detection of high-risk prostate cancer.

## 1. Background

Prostate cancer is the most common malignancy among men in the UK and remains a significant public health concern, accounting for substantial morbidity and mortality [1]. Despite its clinical impact, the UK lacks a national prostate cancer screening program, leading to inconsistencies in early detection and disparities in access to testing [2]. Unlike some European countries that have started pilot studies evaluating the implementation of a risk-stratified population-based approach to prostate cancer screening [3], prostate-specific antigen (PSA) testing in the UK remains largely opportunistic, depending on primary care practices or community-led programs [4]. This fragmented approach has contributed to variations in prostate cancer diagnosis and potential delays in early detection.

PSA testing has been a cornerstone of early prostate cancer detection since its introduction in the 1970s [5,6]. However, concerns over its potential to over-diagnose indolent tumours, which leads to overtreatment, have fuelled debates about its role in routine screening [7]. Recent advances in risk stratification, including the incorporation of additional biomarkers and predictive models, have renewed interest in PSA-based screening as part of a personalised approach to prostate cancer detection [8]. Risk assessment tools such as the Riskman score, which integrates PSA, the free-to-total PSA ratio (PSAft%), and age, offer promising alternatives for improving risk classification and reducing unnecessary interventions [8].

This study evaluates the performance and outcomes of the Graham Fulford Charitable Trust (GFCT)’s community-based PSA testing program by analysing historical testing records and participant feedback. Founded in 2004 by Graham Fulford, the GFCT was established to raise awareness of prostate cancer following the loss of two close friends to the disease and the absence of a national screening program. Since then, it has grown considerably, offering a widescale testing program that has earned a Kings Award for community service for its founder. We evaluate the predictive performance of the GFCT traffic light scoring system for stratifying prostate cancer risk, using the previously developed Riskman score as a benchmark. While Riskman provides a useful point of comparison, the primary focus of this study is the real-world evaluation of the GFCT traffic light system. In addition, we explore ongoing developments in PSA screening strategies across Europe and the USA.

## 2. Materials and Methods

### 2.1. Data Collection

Routine data have been collected on the number of tests and the characteristics of the men since the charity began. Data on the age, PSA, and results in terms of classification using the traffic light system have also been collected.

Given that the GFCT does not routinely have access to clinical outcomes, as the men are provided with their results to inform them on whether they may want to seek further testing within the NHS, the charity sought to obtain data on the outcomes as part of an online survey hosted by the GFCT between August 2021 and September 2024. The time elapsed between the PSA test and the completion of the questionnaire was six months. A reminder was sent out 3 months after no response from the first invitation. A further reminder was sent out 2 weeks later. The survey comprised 13 items assessing demographics, PSA levels, family history of prostate cancer, IPSS scores, and self-reported health outcomes. The International Prostate Symptom Score (IPSS) is a questionnaire used to assess the severity of urinary symptoms in men [9]. Participants provided informed consent before completing the survey. Although the questionnaire was designed to capture clinically relevant variables, it was not formally validated. 

### 2.2. PSA Risk Stratification

The PSA results were categorised into three risk levels: green (low risk), amber (intermediate risk), and red (high risk). Risk categorisation was adapted from age-specific PSA thresholds (Table 1) [10]. The Riskman algorithm was employed as a benchmark, with its performance evaluated independently of this study and reported in a separate publication by the lead author [8]. The algorithm is informed by prior work from the Sunnybrook group in Canada, which highlighted the value of incorporating the percentage of free-to-total prostate-specific antigen (PSAft%) into risk prediction models [11].

The Graham Fulford Charitable Trust (GFCT) clinical advisor had previously recommended the routine collection of PSAft% data for many years, enabling its inclusion in this analysis. The primary focus here was to assess the performance of the GFCT traffic light system. The Riskman algorithm is used here as a benchmark to compare how it performs when applied to the GFCT dataset. No model re-training or recalibration was conducted.

Another reason for evaluating the Riskman score in this context is that, while the GFCT traffic light system is based solely on PSA values, the GFCT has routinely collected free-to-total PSA ratios (PSAft%). However, this additional data have not been integrated into a formal combined risk score within their framework. Riskman offers an advantage on whether incorporating these routinely collected metrics could enhance risk stratification beyond the PSA-only approach. This analysis helps determine whether a multivariable model could be useful for refining the current system going forward.

### 2.3. Outcome Data

Outcomes were reported according to the self-reported prostate cancer grade group, which was defined using the Gleason score as follows: Gleason score 6 (3 + 3) = prostate cancer grade group 1; Gleason score 7 (3 + 4) = prostate cancer grade group 2; Gleason score 7 (4 + 3) = prostate cancer grade group 3; Gleason score 8 (4 + 4, 3 + 5, and 5 + 3) = prostate cancer grade group 4; Gleason scores 9–10 (4 + 5, 5 + 4, and 5 + 5) = prostate cancer grade group 5.

### 2.4. Data Analysis

Descriptive Analysis

Descriptive statistics were calculated to summarise the participant characteristics. To visualise the geographic coverage, a map of England was generated using approximately 75,000 postcodes of the participants in the last 5 years. Postcodes were geocoded to obtain geographic coordinates (latitude and longitude), enabling spatial visualisation. Microsoft Excel was used to plot the distribution of these coordinates.

Statistical Analysis

To examine the associations between clinical variables and prostate cancer risk, univariate and multivariable logistic regression analyses were performed. The predictor variables included age, PSA (prostate-specific antigen), and PSA free-to-total ratio (PSAft%), all treated as continuous variables. The assumption of linearity between continuous predictors and the log-odds of the outcome was assessed graphically using LOWESS (Locally Weighted Scatterplot Smoothing) curves. The LOWESS plots indicated approximately linear relationships for age and PSA. However, the relationship between PSAft% and prostate cancer risk was non-linear; thus, a log transformation was applied to PSAft% prior to inclusion in the regression models. Odds ratios (ORs) with 95% confidence intervals (CIs) were reported.

Risk Stratification

To assess the GFCT “traffic light” risk stratification approach (categorised as green, amber, and red based on accepted PSA thresholds), an age-adjusted logistic regression was conducted. The outcome variables were any prostate cancer (Grade Group ≥ 1) and clinically significant prostate cancer (Grade Group ≥ 3). The traffic light risk categories served as the primary predictor variable.

The comparative performance of the Riskman Score and GFCT risk stratification approach was evaluated for prostate cancer Grade Group ≥ 1 and Grade Group ≥ 3 (clinically significant prostate cancer) using the area under the receiver operating characteristic curve (AUC).

Statistical analyses were performed using STATA 17 [12].

## 3. Results

A total of 4847 men who participated in community PSA testing events were invited to complete the survey, with 2422 responding (response rate: 50%).

Table 2 shows the number of PSA tests carried out since 2004 by age group. Together with their partners, the GFCT has been involved in testing over 131,873 men and conducting nearly 327,129 tests. Figure 1 shows the geographic distribution of men who underwent PSA testing with the GFCT, based on residential postcodes.

Table 3 shows the general characteristics of the respondents (N = 2422) and summarises the data characteristics by self-reported prostate cancer status. The study cohort had a mean age of 65 years, with 98% identifying as Caucasian. The prostate cancer group was slightly older (65.6 years vs. 64.7 years), but the difference was minimal. The prostate cancer group had significantly higher PSA levels (mean 13.9 ng/mL vs. 5.8 ng/mL), whilst the mean of PSAft% was not different between the two groups.

### 3.1. Risk Group Analysis

Table 4 shows the PSA statistics categorised by PSA results in the form of traffic light indicators, with a total number of 2048 participants. Significant differences in the PSA statistics were observed across the green, amber, and red categories (*p*-value < 0.001). Higher PSA levels were consistently associated with increased prostate cancer grade groups, as demonstrated in Table 5.

### 3.2. PSA Risk Stratification (As Green, Amber, and Red) and Riskman Score Performance

Table 6 shows risk estimates for the GFCT risk stratification in relation to prostate cancer of Grade Group ≥ 1 (any cancer) and Grade Group ≥ 3 (clinically significant cancer) and the previously defined Riskman score based on age, PSA, and PSAft%. The GFCT risk stratification approach demonstrated that individuals classified in the “red” risk group had markedly increased odds of harbouring any prostate cancer (OR: 16.545; 95% CI: 7.723–35.442) and clinically significant cancer (OR: 15.222; 95% CI: 4.804–48.228) compared to the “green” reference group. The “amber” category was associated with increased odds for Grade Group ≥ 1 (OR: 2.757; 95% CI: 1.233–6.166) but not significantly so for Grade Group ≥ 3 (OR: 1.745; 95% CI: 0.497–6.133). Within the Riskman approach, prostate-specific antigen (PSA) was a significant predictor of both any prostate cancer (OR: 1.153; 95% CI: 1.124–1.183) and clinically significant disease (OR: 1.158; 95% CI: 1.117–1.200), while a lower percentage of PSAft% was associated with a reduced likelihood of both outcomes (Grade Group ≥ 1: OR: 0.585; 95% CI: 0.492–0.697; Grade Group ≥ 3: OR: 0.467; 95% CI: 0.310–0.703). Age was not a significant predictor in either case.

Table 7 shows the area under the receiver operating characteristic curve (AUC) values of the GFCT risk stratification and Riskman approaches for predicting prostate cancer of Grade Group ≥ 1 and Grade Group ≥ 3. For the detection of any prostate cancer (Grade Group ≥ 1), the GFCT risk stratification achieved an AUC of 0.72 (95% CI: 0.70–0.74), while the Riskman model demonstrated a higher discriminative ability with an AUC of 0.76 (95% CI: 0.74–0.79). For clinically significant prostate cancer (Grade Group ≥ 3), the GFCT risk stratification approach yielded an AUC of 0.76 (95% CI: 0.73–0.79), whereas the Riskman model again outperformed it with an AUC of 0.84 (95% CI: 0.80–0.88). These results indicate that, while both models offer useful discrimination, the Riskman approach provides greater predictive accuracy, particularly for identifying clinically significant prostate cancer.

Figure 2 show ROC curves illustrating the predictive performance of the GFCT risk stratification approaches and the Riskman Score. Figure 2a presents the ROC curve for the GFCT risk stratification approach for Grade Group ≥ 1 and Figure 2b for Grade Group ≥ 3. Figure 2c shows the ROC curve for the Riskman Score in predicting prostate cancer for Grade Group ≥ 1 and Figure 2d for Grade Group ≥ 3 (clinically significant prostate cancer). The curves demonstrate that the Riskman Score achieves higher discriminative accuracy compared to the GFCT for clinically significant disease.

## 4. Discussion

Men’s health screenings continue to lag behind women’s, particularly for prostate cancer, which remains the most common cancer in men but lacks a national screening program. Additionally, men are less likely to engage in routine health checks, leading to preventable deaths from cancers and chronic diseases such as diabetes and cardiovascular disease [13]. Despite advances in cancer diagnostics and treatment, disparities persist, emphasising the need for proactive screening initiatives.

Main Findings

The GFCT organisation has worked alongside partner charities and men’s health groups to facilitate testing for over 131,873 men, conducting nearly 327,129 tests. This initiative has led to the identification of more than 3348 cases of prostate cancer, many of which might have otherwise gone undetected. The GFCT’s community-based PSA testing program (https://gfct.mypsatests.org.uk/ (accessed on 15 April 2025)) has achieved widespread uptake and addresses a significant gap in prostate cancer testing in the UK, as evidenced by the distribution of postcodes of men who underwent PSA testing organised by the GFCT.

The frequencies of the men’s PSA level are shown. The red group corresponds to individuals with the highest risk of prostate cancer, the amber group represent an intermediate risk level, and the green group corresponds to individuals with the lowest risk.

Our results also suggested a clear and significant relationship between PSA levels and prostate cancer grade groups. Specifically, the mean PSA levels increase progressively with higher grade groups, which typically indicate more advanced or aggressive forms of prostate cancer.

Notably in multivariable logistic regression, the PSA levels were found to be a significant predictor of prostate cancer risk. Specifically, a 15% increase in the likelihood of developing prostate cancer was observed with one unit of elevated PSA levels (1 ng/mL). This result is consistent with the existing literature, which has long established PSA as a critical biomarker in the early detection and monitoring of prostate cancer [14].

Conversely, the study found that one percentage increase of PSAft% was associated with a substantial reduction in prostate cancer risk, approximately 40%. In other words, if the free PSA is higher than the total PSA, the ratio will be high, suggesting a lower risk of prostate cancer. Conversely, if the amount of free PSA in the blood is lower than the total PSA, the free-to-total ratio will be low, indicating a higher risk of prostate cancer. This inverse relationship suggests that a higher proportion of free PSA relative to total PSA might be indicative of a lower likelihood of malignancy. This finding aligns with previous research suggesting that PSAft% can be a useful adjunct in distinguishing between benign prostatic conditions and prostate cancer, potentially reducing unnecessary biopsies in clinical practice [15,16]. Interestingly, age did not emerge as a significant factor in predicting prostate cancer risk in this study. While age is generally considered a major risk factor for prostate cancer, this result may reflect the characteristics of the study population.

Beyond PSA testing, the GFCT has collected data to facilitate the use of the Riskman score, an inexpensive tool that integrates clinical variables such as PSA levels, PSAft%, and age, improving predictive accuracy compared to PSA testing alone. The GFCT risk stratification model, although simpler and categorical, effectively identified individuals at elevated risk. Men classified in the “red” group had more than 15-fold increased odds of clinically significant prostate cancer compared to those in the “green” group. However, the “amber” category showed inconsistent associations, particularly for high-grade disease, indicating potential limitations in the intermediate risk classification (insufficient power).

In the logistic regression models, PSA and PSAft% were consistently strong and statistically significant predictors of the Riskman score for both outcomes. Notably, a lower PSAft% was associated with markedly increased odds of cancer, reinforcing its value as a discriminative biomarker in risk-based models.

ROC analysis further highlighted the enhanced performance of the Riskman score. The AUC for predicting Grade Group ≥ 3 was 0.84 for Riskman, compared to 0.76 for the GFCT, indicating a substantial improvement in discriminative accuracy. While both models showed reasonable performance for detecting any cancer (Grade Group ≥ 1), Riskman again performed better (AUC 0.77 vs. 0.72). These findings suggest that multivariable models incorporating continuous clinical parameters may offer more refined risk stratification than categorical frameworks.

The clinical implications of these results are two-fold. First, they support the use of the Riskman score as a robust decision support tool for the early identification of men at risk of clinically significant prostate cancer. Second, while the GFCT provides a more accessible and interpretable framework, particularly for primary care or community-based settings, its predictive accuracy—especially in the intermediate risk range—may be insufficient for guiding biopsy decisions alone.

Although not developed within this study, the Riskman score was applied as a comparator tool, while the main novel findings pertained to the performance of the GFCT traffic light stratification system.

These findings highlight that the GFCT tool, while simpler and more cost-effective than many complex predictive models, delivers competitive performance based on the self-reported data collected in the survey. Several key features contribute to its success. First, the tool employs age-specific PSA thresholds, which help account for natural increases in the PSA levels with age. This adjustment enhances the signal-to-noise ratio, allowing the model to more accurately identify abnormal results. Second, it classifies individuals into discrete risk groups, providing clear and actionable categories that are easy for both participants and clinicians to interpret and act upon. It should be noted, however, that the study cohort—comprised of survey responders—was likely enriched with individuals at a higher pre-test risk of prostate cancer. This enrichment may have amplified the apparent performance of the model. Finally, while not formally integrated into the core model, additional PSA-related metrics such as PSAft% and reflex testing are routinely collected in the GFCT setting. Their informal use may enhance triage decisions, further supporting the tool’s practical utility. When compared to the integrated assessment—Riskman—the GFCT traffic light system showed reasonably good agreement but slightly lower performance on detecting clinically significant cancers.

Strengths and Limitations

The key strengths of this initiative include its large-scale accessibility, focus on community engagement, and adaptability to evolving screening strategies. By integrating a tiered risk-based approach, the GFCT has provided more personalised and effective screening options for men at risk of prostate cancer. However, limitations exist, as the reliance on self-reported data introduces potential biases, and the predominantly Caucasian cohort limits generalisability. A further possible bias is due to a response rate of just above 50%. Efforts to engage underrepresented populations, particularly Black men and those with a family history of prostate cancer, are essential in ensuring equitable access to screening [17]. Studies have shown that Black men are at a higher risk of developing aggressive prostate cancer [18], making targeted outreach crucial. Additionally, the variability in PSA cut-off points across different screening guidelines poses challenges in standardisation, necessitating ongoing research and policy adjustments [19]. As the time lag between the PSA test and survey completion was approximately six months, this may introduce recall bias. Furthermore, recall bias is likely to be lower among men subsequently diagnosed with prostate cancer. In terms of missing data, the non-prostate cancer group has a higher proportion of missing values compared to the prostate cancer group, which may indicate recall bias introduced by being non-cases. This suggests that individuals in the non-prostate cancer group may be less likely to recall or report certain information compared to those diagnosed with prostate cancer, possibly due to differences in attention to health details.

The prediction model using logistic regression has several limitations that may impact its reliability and clinical applicability [20]. Overfitting is a concern, especially if too many predictors are included, which may reduce the generalisability. Missing data in key variables, along with potential recall bias, particularly in the non-prostate cancer group, could introduce systematic errors. Additionally, the assumption of linearity in the log-odds may not hold for all continuous variables, as seen with PSAft%, requiring transformations. The GFCT risk stratification approach offers several practical strengths in community-based screening settings. Its intuitive traffic light system (green, amber, and red) provides a clear, accessible framework that facilitates communication and supports shared decision-making, particularly among individuals unfamiliar with PSA interpretation. The approach effectively identifies individuals at highest risk, with those in the “red” category showing a markedly increased likelihood of clinically significant prostate cancer. This simplicity supports broad implementation in non-specialist settings and empowers participants to engage in informed follow-up discussions with healthcare providers.

However, the approach also has notable limitations. As a categorical model, it lacks the granularity of continuous risk predictors and may oversimplify the risk, particularly in the “amber” group, which demonstrated limited discriminatory power for high-grade disease. This may result in uncertainty in clinical decision-making for men falling into this intermediate-risk category. Additionally, the GFCT model does not currently incorporate additional biomarkers, imaging findings, or comorbidity factors, which may enhance risk stratification. While its accessibility is a key strength, future integration with validated multivariable tools like the Riskman score may improve its predictive accuracy and clinical utility without compromising its usability in community settings.

Furthermore, whilst the GFCT tool demonstrates good performance on the survey dataset, it is important to acknowledge that reliance on self-reported data may introduce some inflation in risk categorisation, particularly in the red and amber groups. Self-report measures can be subject to recall bias or overreporting, potentially leading to an overestimation of individual risk. Nevertheless, when comparing model performance, the overall risk score achieved by the Riskman tool remains broadly consistent with the findings from other validated approaches. Notably, its predictive strength aligns closely with the results of Lophatananon et al. [8] in the TARGET study, which assessed the efficacy of various biomarkers and endpoints in refining referrals for suspected prostate cancer. This convergence in findings suggests that, despite methodological differences—including reliance on self-reported vs. clinically collected data—the GFCT approach yields comparable risk stratification to more complex and biomarker-intensive models.

Comparison with Other Studies

Internationally, the UK, Europe, and the US have adopted different prostate cancer screening strategies. The UK currently lacks routine screening but relies on PSA testing upon request. The TRANSFORM trial (EUR 42 million investment) aims to determine the most effective approach for national implementation [21]. However, as this study may take up to 20 years to produce survival outcome data, the GFCT remains crucial in providing immediate testing for men at risk.

In Europe, there is a shift toward risk-based screening to reduce overtreatment [19]. Several countries are now running pilot programs that refine screening strategies based on stratifying follow-up periods according to PSA levels. Some European models incorporate MRI as a primary triage tool, improving specificity in identifying clinically significant cancers [22]. Meanwhile, the US is piloting genetic-based approaches such as the ProGRESS study in the Veterans Program, which tailors screening recommendations based on genetic risk [23]. The increasing use of PRS and MRI in screening programs demonstrates a growing shift toward personalised cancer detection strategies [24,25].

Moreover, other large-scale efforts, such as the Stockholm3 study, have demonstrated that combining genetic profiling with traditional biomarkers improves the risk assessment [26]. Integrating multi-modal approaches, including imaging and biomarkers, has also been a focus of recent studies, highlighting the need for individualised screening strategies [27,28].

GFCT and PSA Testing: A Comparative Overview

Unlike prostate-specific antigen (PSA) testing conducted within the NHS, GFCT-facilitated testing operates independently of the national health system. In the GFCT model, men receive their PSA results directly and, if the result is elevated, are advised to consult their general practitioner (GP). While it is often stated that the GFCT does not provide pre-test counselling, this is not strictly true. GFCT counselling is delivered in the form of online fact sheets and written information rather than through the face-to-face discussions typically offered by GPs within the NHS. This distinction in delivery format may lead to differences in how informed consent is perceived.

Receiving PSA results outside of NHS systems can result in a period of uncertainty or anxiety and, at times, create friction with GPs who were not involved in the initial testing decision. However, in practice, most GPs respond constructively. When presented with an elevated PSA result from a charity-led screening, they frequently repeat the test via NHS pathways for confirmation.

Challenges occasionally arise when men also present results from more advanced reflex tests, such as Stockholm3 or Proclarix, which may be offered through the GFCT testing pathway. These tests are not yet standard within the NHS, and some GPs may be unfamiliar with their interpretation. Responses vary depending on the GP’s familiarity with prostate cancer diagnostics. Those with relevant expertise often incorporate the additional data into their clinical assessment, while others may feel uncertain or sidelined by results that do not align with conventional NHS protocols. It is important to note, however, that GFCT adheres to NICE guidelines, and the test results provided fall within accepted diagnostic frameworks. Despite this, some GPs do not act on the results, underscoring the ongoing variability in clinical practice.

This dynamic reflects broader uncertainty within the prostate cancer diagnostic landscape. Despite progress over the past two decades, the NHS still lacks a nationally coordinated strategy that balances men’s expectations, clinical best practices, and cost-effectiveness. As a result, charity-led initiatives like those offered by the GFCT have emerged as important contributors, responding to the growing demand from men for accessible and proactive health engagement.

A key priority moving forward is the optimisation of shared decision-making within these community-based screening programmes. The GFCT supports informed choice through a combination of written materials, a comprehensive website, access to online consultations with medical advisors, and—starting 1st May 2025—a free doctor’s consultation for men whose GP does not offer a clear clinical pathway. Perhaps the most important strength of these initiatives is the way they empower men to take greater ownership of their health.

Historically, men have often delayed help-seeking or ignored symptoms, a pattern reinforced by the NHS’s largely reactive model of care. The GFCT’s growing portfolio of health checks seeks to address this by encouraging earlier engagement, reframing preventive care as an accessible and actionable priority, and promoting a culture of self-responsibility and timely intervention.

Implications

The GFCT’s model offers a scalable and inexpensive solution for improving prostate cancer testing in the UK. The charity-led initiative has demonstrated that community-based programs can effectively bridge gaps in healthcare access and contribute to early detection. Furthermore, collaborations with national screening trials, such as TRANSFORM, could enhance the development of an evidence-based, risk-stratified screening program. The success of the GFCT highlights the potential for integrating community-based programs into national healthcare frameworks.

The integration of advanced diagnostic tools, including MRI and biomarker testing, has the potential to improve screening specificity and reduce unnecessary biopsies [22]. Emerging technologies, such as genetically adjusted PSA (gaPSA) scores, can refine PSA thresholds based on an individual’s genetic background, enhancing precision [25,29]. Additionally, expanding the GFCT’s scope to include other chronic disease screenings (e.g., cardiovascular disease, diabetes, and kidney function) could encourage broader engagement with preventive healthcare. Addressing multiple health concerns simultaneously may increase participation in health monitoring programs and reduce the burden of chronic disease among men.

Future directions should also consider developing a hybrid model or a tiered approach that combines PSA-based risk stratification with genetic screening, MRI, and other biomarkers to create an efficient, personalised pathway for early prostate cancer detection. Research into how risk-stratified follow-up intervals can be optimised would further improve the efficiency of screening programs [30].

## 5. Conclusions

The GFCT plays an important role in PSA screening across the UK, offering a community-based alternative in the absence of a national screening program or widespread NHS access to PSA testing. By empowering men to take proactive steps toward early detection, the initiative helps bridge critical screening gaps. As the program continues to expand, it has the potential to further improve early diagnosis rates and reduce disparities in prostate cancer detection, contributing to more effective long-term cancer control efforts.

The GFCT’s traffic light system, based on widely used PSA thresholds, demonstrated good predictive performance for detecting any prostate cancer, reinforcing its value as an effective tool for risk communication and clinical triage. However, the Riskman score showed superior accuracy in identifying clinically significant cancers. Given its simplicity, low cost, and improved discrimination, the Riskman score represents a practical enhancement to the current screening strategies and is well suited for widespread implementation in future prostate cancer testing programs.

The GFCT’s risk-based prostate cancer testing program is a widely accessible and adaptable initiative that could serve as a model for future screening programs in the UK. By incorporating PSA testing, genetic risk assessment, and MRI imaging, this hybrid screening strategy presents a promising approach to optimising early detection while minimising unnecessary interventions. The integration of PRS and biomarker-based risk stratification represents the next step in refining prostate cancer screening, aligning with emerging trends in precision medicine.

Continued research is necessary to refine screening algorithms and improve equity in prostate cancer care. Addressing the disparities in access to screening, particularly among high-risk groups, remains a priority. Black men, in particular, face disproportionate risks and require targeted interventions to ensure better outcomes. By supporting ongoing trials, real-world evidence, and innovative research, these initiatives have the potential to transform prostate cancer screening and management nationwide, ultimately reducing mortality and improving patient outcomes. Strengthening collaborations between charities, healthcare providers, and policymakers will be crucial in ensuring that prostate cancer screening evolves to meet the needs of all men, regardless of their background. 

## Figures and Tables

**Figure 1 cancers-17-01728-f001:**
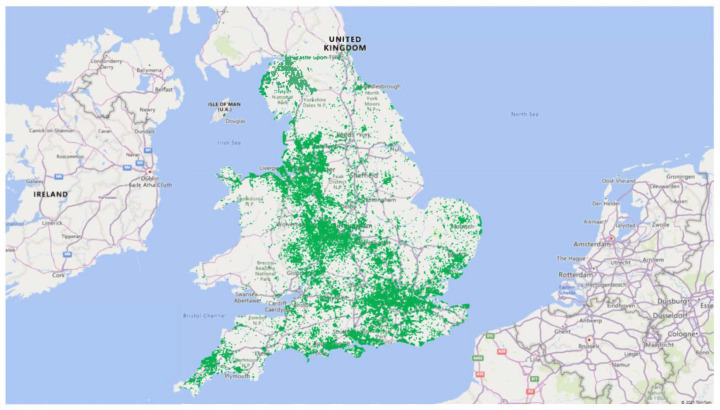
Map depicting the distribution of men who underwent PSA testing with the GFCT in the last 5 years.

**Figure 2 cancers-17-01728-f002:**
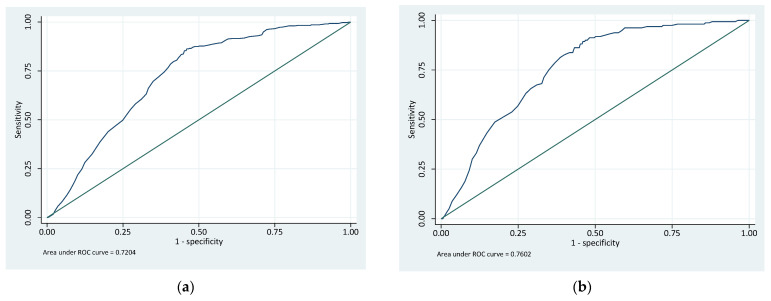
(**a**) ROC of the GFCT risk stratification for prostate cancer Grade Group ≥ 1 and (**b**) for prostate cancer Grade Group ≥ 3. (**c**) ROC of the Riskman score for prostate cancer Grade Group ≥ 1 and (**d**) for prostate cancer Grade Group ≥ 3.

**Table 1 cancers-17-01728-t001:** PSA age-specific cut point used by the GFCT.

Age	<1.50	1.50–2.49	2.50–2.99	3.00–3.99	>4.00
40–49					
50–69					
70 and over					

Follow-up recommendation: Those that receive a “green” letter are recommended for a 3-year follow-up. Those that receive an “amber” letter are recommended for a 2-year follow-up. Those that receive a “red” letter are recommended for a 1-year follow-up.

**Table 2 cancers-17-01728-t002:** Number of PSA tests by age.

Age	Highest PSA(ng/mL)	Average PSA(ng/mL)	No of Tests
<30	6.6	0.79	217
30–39	20.6	0.81	1775
40–49	72.7	0.9	43,407
50–59	952	1.31	99,197
60–69	1803	2.1	104,277
70–79	12,463	3.04	68,584
80–89	361	3.44	9441
90–99	139	4.59	229
>100	1.6	1.55	2
All ages	-	-	327,129

**Table 3 cancers-17-01728-t003:** General characteristics by self-reported prostate cancer status.

Variables	Non-Prostate Cancer Group (N (%))	Prostate Cancer Group (N (%))
	No = 1946	No = 476
Age (years old)	range:34.0–90.0	range:43.0–84.0
	mean 64.7 SD * (8.9)	mean 65.6 SD * (7.9)
Unknown	309 (15.9%)	59 (12.4%)
PSA (ng/mL)	range:0.0–260.3	range:0.0–242.0
	mean 5.8 SD * (8.0)	mean 13.9 SD * (23.4)
Unknown	302 (15.5%)	58 (12.2%)
PSAft%	range: 0.0–22.8	range: 0.0–9.6
	median 0.78 IQR ** (0.43–1.17)	median 0.77 IQR ** (0.29–1.35)
	433 (22.3%)	63 (13.2%)
Ethnicity		
African-Caribbean	7 (0.4%)	2 (0.4%)
Asian	5 (0.3%)	1 (0.2%)
Mixed Race	8 (0.5%)	1 (0.2%)
Other	7 (0.4%)	1 (0.2%)
White European	1551 (93.7%)	445 (93.5%)
White Other	77 (4.7%)	26 (5.5%)
Unknown	291 (15%)	0 (0%)
IPSS		
Mild	615 (47.5%)	152 (43.8%)
Moderate	578 (44.6%)	165 (47.6%)
Severe	102 (7.9%)	30 (8.6%)
Unknown	651 (33.5%)	129 (27.1%)
Family history of prostate cancer		
No	1489 (95.4%)	424 (94.9%)
Yes	72 (4.6%)	23 (5.1%)
Unknown	385 (19.8%)	29 (6.1%)

* Standard deviation. ** Interquartile range.

**Table 4 cancers-17-01728-t004:** PSA statistics according to traffic light groups.

Group	Numbers	Mean PSA (ng/mL)	SD *	Min.	Max.
Green	253	1.72	1.15	0.03	5.65
Amber	680	4.47	1.37	2.00	17.90
Red	1115	10.56	17.14	2.53	260.29

* Standard deviation. F-test *p*-value 0.0001. There is statistical difference between the means of the 3 groups.

**Table 5 cancers-17-01728-t005:** Mean PSA according to prostate cancer grade groups.

Prostate Cancer Grade Group	No (%)	Mean PSA (ng/mL)	SD *
No cancer	1644 (79.7)	5.8	8.0
Group 1	73(3.5)	8.8	9.9
Group 2	186 (9.0)	10.8	18.6
Group 3	78 (3.8)	13.3	14.9
Group 4	38 (1.8)	23.9	36.0
Group 5	43 (2.1)	28.1	42.4
Total	2062	7.4	13.1

* Standard deviation.

**Table 6 cancers-17-01728-t006:** Estimates of prostate cancer risk by age, PSA, and PSAft% in a multivariable logistic regression model.

Approach	Prostate Cancer Grade Group	Variables	Odd Ratios	95% C.I.	
				Lower	Upper
GFCT risk stratification	Grade Group ≥ 1	Age	1.013	0.999	1.026
		Green (Reference)	1.000		
		Amber	2.757	1.233	6.166
		Red	16.545	7.723	35.442
		Total no	2050		
	Grade Group ≥ 3	Age	1.038	1.017	1.059
		Green (Reference)	1.000		
		Amber	1.745	0.497	6.133
		Red	15.222	4.804	48.228
		Total no	1794		
Riskman score	Grade Group ≥ 1	Age	0.997	0.983	1.011
		PSA	1.153	1.124	1.183
		PSAft% *	0.585	0.492	0.697
		Total no	1920		
	Grade Group ≥ 3	Age	1.024	0.998	1.050
		PSA	1.158	1.117	1.200
		PSAft% *	0.467	0.310	0.703
		Total no	1371		

* Log-transformed PSAft%.

**Table 7 cancers-17-01728-t007:** Area under the curve (AUC) by the GFCT risk stratification and Riskman.

Approach	Prostate Cancer Grade Group ≥ 1			Prostate Cancer Grade Group ≥ 3		
	AUC	95%C.I		AUC	95%C.I	
		Lower	Upper		Lower	Upper
GFCT risk stratification	0.72	0.70	0.74	0.76	0.73	0.79
Riskman	0.76	0.74	0.79	0.84	0.80	0.88

## Data Availability

Data are available upon request directly to the Graham Fulford Charitable Trust (e-mail: info@psatests.org.uk).

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
