# Peer review of "Charity-Provided Community-Based PSA Testing for Assessment of Prostate Cancer Risk in the UK: Clinical Implications and Future Opportunities"

_cancers, 2025, doi:10.3390/cancers17101728_

Round 1

Reviewer 1 Report

Comments and Suggestions for Authors

This paper presents the results of a community-based PSA testing initiative run by a charity in the UK. The GFCT uses an online questionnaire to identify high-risk patients based on simple criteria such as PSA, age, race, baldness pattern, and finger length, making it a very important study for society. However, there are some insufficient explanations and figures for deriving the results, and it needs to be significantly revised.

1.     The color coding is difficult to understand.

Table 1. PSA age specific cut point. For example, for people aged 40-49, PSA0-0.99 is low risk, PSA1.00-1.49 is intermediate risk, and PSA1.5-2.0 is high risk. Is it correct to color code low risk as green, intermediate as amber, and high risk as red? If so, please make it clear in the Figure. Next, for people aged 40-49, which category do those with a PSA of 2 or higher belong to? The same is true for other age groups, but where do those with a PSA value above Max PSA belong?

2.     Important results are not shown in the Figures or Tables. In the Results section (p.3 l.129-135) you stated “There are no significant differences in the distributions of baldness in the 40s concerning prostate cancer status, IPSS, or PSA results categorised by traffic lights (green, amber, or red) with all chi-square p-values ​​>0.05. Regarding hand patterns, 72% reported a male hand pattern (index finger shorter than the ring finger). This variable also showed no significant differences in distribution between groups based on prostate cancer status, IPSS, and PSA results by traffic light categories (all chi-square p-values ​​>0.05)." I think these results are important in this study, and should be shown in a Figure.

3.     Finally, only age, PSA, and PSAf/t ratio were analyzed in the multivariate analysis. The area under the ROC curve was only 0.77, which is not a very good value and unfortunately lacks the new information provided by this screening test.

4.     The explanation of the charity provided community based initiative that follows the study limitation on page 9 seems too long and has little relevance to the results of this study.

Author Response

We are grateful for the reviewers' comments. As noted by other reviewers as well, we have made major revisions to our manuscript. 

We accept the need for significant revision of the manuscript which we have done in the new version. We have also removed some of the data fields that did not reach statistically significant to make the tables clearer as well.

We think that this has resulted in a simpler and clearer paper in keeping with all the reviewers’ comments.

Reviewer 2 Report

Comments and Suggestions for Authors

The subject of community-based PSA testing in the UK is important.

The manuscript must be strengthened by rewriting it in a more concise and precise manner. I also recommend adopting a different statistical methodology to better address several associations identified in the study.

Below are my specific questions and suggestions.

Abstract

1.       Define the abbreviations "PSA" and "IPSS," as the journal’s audience may include readers beyond oncologists and urologists.

2.       Line 17: Correct "particpants" to "participants."

3.       Objective and Line 22: The manuscript does not report any data on participants' attitudes to the service.

4.       Line 23: The statement "but not with higher IPSS categories" contradicts the conclusion in Line 198.

5.       Line 24: Use consistent terminology for "free-to-total PSA ratio" throughout the manuscript. Different notations appear e.g. in Lines 24, 100, 121, and 127.

6.       Line 25: No evidence is provided to support the claim that PSA alone is a poorer predictor compared to combinations with other markers.

Introduction

7.       Compare prostate cancer mortality and stage at diagnosis in the UK with EU countries.

8.       Define abbreviations PSA, GFCT, and NHS.

9.       Lines 45, 65: Move web links to the References section.

10.   Line 50: Insert a comma before "so many."

11.   Lines 71–74: The claim about the program's credibility needs references.

12.   Lines 76–77: Provide references to support the claim of "Riskman" being highly cost-effective.

13.   Line 79: Correct "findingsfrom" to "findings from."

14.   Lines 79–81: Be more specific in describing the study’s objective.

Methods

15. Create a separate section for "Data collection."
16.
Clarify how PSA test results were used to guide men on further actions, inform cancer diagnoses, and specify the instructions provided to them based on their PSA results.

17. Specify where and how final cancer diagnoses were established.
18. Include the questionnaire in an appendix.
19. Indicate the time elapsed between PSA testing and completing the questionnaire.
20. Detail the follow-up period for cancer outcomes after PSA testing.
21. Explain how the "colour" groups were constructed (Table 1 needs an explanation).
22. In Table 1, explicitly mention the "colour" groups.
23. Lines 90–91: Move information on the number of patients to the Results section.
25. Shift the description of data analysis (Lines 99–102) to the "Data analysis" section.
26. Line 101: Remove "with these 3 variables that the GFCT routinely collect."
27. State that patients were also asked about free-to-total PSA ratio, symptoms, and ethnicity.

 Data Analysis

28. Mention "Riskman" explicitly in this section.
29. Clarify whether Age, PSA, and PSAft were introduced into the logistic regression model as categorical or continuous variables. If continuous linear, describe how a linear relationship between an independent variable and log odds cancer was checked.
30. If only one logistic regression was performed, use "multivariable logistic regression."
31. You based all your conclusions (except for the relationship between markers and cancer) solely on p-values. This is not considered good practice and may be misleading. It is preferable to calculate the estimates of differences and their confidence intervals, as these provide much more informative results. Please present the results as estimates of difference along with their confidence intervals (CIs) instead of relying on p-values. This may require using alternative statistical tests (e.g., linear or logistic regression instead of ANOVA and chi-squared tests) and reorganizing the Results section accordingly.
32. Indicate whether measurements were unique per participant or repeated. If the latter, explain how analyses accounted for this.
33. It is unclear whether the assumptions for the ANOVA test were met. (This may become irrelevant if other tests (se comment 31) are used instead).
34. I didn’t find any results based on t-tests in Results.
35. Line 114: Replace "Chi-square" with "Pearson’s chi-square."

Results
36. Report the number and percentage of men invited, responded, and included in the main analysis.
37. Round percentages to one decimal place.
38. Define abbreviations in all tables and figures.
39. Lines 126–127: Provide units of measurement for PSA results.
40. Table 2: Present characteristics stratified by cancer and non-cancer groups.
41. Table 2: Revise the title to "General Characteristics of 2,422 Responding Men."
42. Table 2: Combine columns 2 and 3.
43. Table 2: Use "N" instead of "no".
44. Table 2: State how many men had missing PSA and PSAft% results.
45. Table 2: Add "Unknown" as a category for variables with missing data.
46. Table 2: Specify units of measurement for continuous variables.
47. Ensure consistent PSA units across all tables and text (se e.g., Tables 1-2, line 162).
48. Table: Correct "Hand pettern 1" to "Hand pattern 1".
49. Lines 129–131, 133–135: Combine these sentences for brevity.
50. Line 140: Use "<0.01" instead of "<0.05."
51. Table 3A: Label "group" as e.g. "PSA group."
52. Standardize notations such as "N" and "SD" across tables.
53. Table 3A: Replace "0.0000" with "<0.01" in Table 3A.
54. Consider omitting Table 3B.
55. Line 150: Use "p-value=0.13" instead of "Chi-square p-value > 0.05."
56. Line 152: Use "p-value=0.04" instead of "p-value < 0.05."
57. Revise the title of Table 4 to e.g. "Distribution of IPSS categories across PSA groups."
58. Combine percentages with numbers in Table 4 (e.g., "Mild, n (%)").
59. Tables 5 and 6 can be reorganized and merged with Table 3.
60. Line 162: Change "risk estimates" to "estimated odds ratios of cancer."
61. Revise Table 7’s title to e.g. "Estimated Odds Ratios of cancer for age."
62. Use "multivariable" instead of "multivariate" in Table 7.
63. Remove p-values from Table 7, as CIs provide sufficient information.
64. Line 165: Capitalize "Figure 1."
65. Define abbreviations "ROC" and "AUC."
66. Interpret results from Figure 1.
67. Add a y-axis label to Figure 1.
68. Revise Figure 1’s title for clarity: "Receiver Operating Characteristics (ROC) Curve for prostate cancer model including age, PSA, and PSAft%."

Discussion
69. Shorten the Discussion section to avoid repetition.
70. Begin the Discussion with the most significant findings, followed by secondary results.
71. Avoid referencing p-values, statistical tests, tables and figures in this section.
72. Remove redundant statements (e.g., Lines 172–176 – this information has already been given in Introduction, lines 69-73).
73. Lines 180, 181: Delete “(International Prostate Symptom Score)” and “(Prostate-Specific Antigen)”.

74.   Line 183: Delete “published”.

75.   Lines 183–185: Explain discrepancies between studies.

76.   Lines 187, 188: Delete, as they repeat the previous paragraph.

77.   Avoid contradictory statements about PSA and IPSS relationships (Lines 189–209). First, you state that PSA and IPSS are not related, but later you claim that they are related. Your conclusions are based on p-values from two tests that measure the same relationship using different techniques. The discrepancies may simply be due to the differences in these methods, as one test may be more sensitive in detecting small differences than the other.

78.   Line 210: Delete “presented in Table 3 A”.

79.    Shorten the text in Lines 210–227. Clarify that Table 3A differentiates PSA levels, not cancer risk.

80.   Line 228: Delete “presented in Table 6”.

81.   Line 261: Delete “PSA free-to-total ratio”.

82.   Lines 266-267: This sentence is redundant (see lines 263-264).

83.   Discuss the results from the evaluation of the prediction model.

84.   Address selection bias by including a table (in the Appendix) comparing the characteristics of participants and non-participants.

85.   Reflect on the response rate and the extent of missing data.

86.   Assess potential information bias. (Self-reporting, the time lag between testing and completing the questionnaire, etc.)

87.   Discuss the limitations of the methodology used to evaluate the prediction model.

88.   The text on lines 306-312 should be condensed for clarity and conciseness.

89.   Did patients in this trial benefit from the “Riskman” score (or if they were informed only about a single PSA result)? If yes, how (e.g., at what probability cutpoint for cancer they were advised to proceed with further investigation?)

90.   Will future users of GFCT be provided with their testing results based on the “Riskman” score?

91.   Lines 312-314: A reference in needed. Alternatively, present these results in the Results section.

92.   Line 330: Elaborate on critics of charities.

Conclusion:

93.   Line 353: Which "other markers" are being referred to?

94.   Lines 353-355: No results are presented to confirm that PSA alone was a worse predictor than when combined with other markers.

Author Response

We are very grateful to this reviewer for these detailed comments and also to the other reviewers for their observations.

We accept the need for significant revision of the manuscript which we have done in the new version. We have also removed some of the data fields that did not reach statistically significant to make the tables clearer as well. We have added and international comparison of approaches and shortened the discussion.

As the layout and line numbers of the new version of the paper are completely different from the original version we cannot provide a point by point response to the detailed review bit hope that our new version address in a more general way the comments the reviewer kindly made and we are very grateful for their detailed feedback which was very helpful in how we considered restructuring the manuscript.

We think that this has resulted in a simpler and clearer paper in keeping with all the reviewers’ comments.

Reviewer 3 Report

Comments and Suggestions for Authors

The authors provide an interesting study on the Charity provided community-based  detection of PSA in male patients in the UK, between 2021 and 2024. Prostate cancer screening is not universally available in the UK, and this is the important clinical background of this paper. Community based programs, like the one the Authors present, can be a useful and powerful substitute for government-based policies, when they are not available to the people.

However, there are a few points which must be clarified and better stated, before the paper can be accepted for publication in Cancers. These points are detailed below for the Authors:

TITLE: I would make the title a bit more assertive, and just state

THE IMPORTANCE OF CHARITY PROVIDED COMMUNITY-BASED PSA TESTING FOR ASSESSMENT OF PROSTATE CANCER RISK IN THE UK: CLINICAL IMPLICATIONS AND FUTURE OPPORTUNITIES

ABSTRACT.

Line 23-25: This “RISKMAN score” is a very original piece of information and a useful approach for prostate cancer risk assessment. It is based on PSA, age and free:total PSA ratio. The Authors should stress better this innovative point, and also describe it better in the Results and Discussion sections of the manuscript.

INTRODUCTION

General comment: This study has been carried out in a specific time interval, which is the one of the COVID pandemic years. A comment about this fact should be added by the Authors in the Introduction and the in the Discussion section. The program provided to the patients an online questionnaire, a very successful strategy during the pandemic years. It is well known that many patients did not seek medical health and did not join screening programs during the pandemic years. Men were particularly affected by the pandemic situation as compared to women, and some oncological guidelines related to men’s cancers were modified during the pandemic years in order to regulate access to hospitals and health services for the people and limit the viral spread while assuring patients’ safety (add cit.: Cakir OO, et al. Management of penile cancer patients during the COVID-19 pandemic: An eUROGEN accelerated Delphi consensus study. Urol Oncol. 2021 Mar;39(3):197.e9-197.e17. doi: 10.1016/j.urolonc.2020.12.005). A program like the one presented by the Authors is an example of good response and safe approach to population screening for cancer risk during the pandemic years This is a very interesting point to underline, and one of the strengths of this paper.

Methodology: Page 3, line 106: Descriptive

DISCUSSION

Page 7, line 184: verb missing: Similar findings were reported by other groups.

Page 8, line 239: cancer patients

Page 8, line 262: Conversely if

Page 9, line 308: the Authors should describe better the Riskman Score, which is a very interesting clinical tool, based on easy-to-get data, like PSA, PSA ratio and patients’ age. Where do they explain it, before the Discussion? This score is a really original clinical tool, that could be better stresses and mentioned also in the Results session. If this score is used for the first time in this manuscript, this fact should be stressed by the Authors.

Page 9, line318: Please re-phrase, unclear: “Men are yet see a significant role…”

Page 9, line 323-329: Please add a comment on the pandemic years, and the perfect fit of this program with that particular moment (online questionnaires, possible mail contact).

Page 9, line 342: “finalisse,” please correct

CONCLUSIONS

This study highlights the importance of community-based initiatives like those of the GFCT, particularly in a context where formal prostate cancer screening programs are not yet established. This study also showcases the combination of PSA with PSA ratio and patients’ age, presenting a novel score, the Riskman Score, which significantly enhances the predictive accuracy for prostate cancer risk assessment, suggesting that a multiple parameters approach (markers + clinical characteristics) could be more effective in prostate cancer testing in the community setting, limiting also the risk of unnecessary biopsies. Finally, this study provides a suitable strategy, like easily available online questionnaires or regular mail services, to safely engage men in their health assessment at critical times, like the pandemic years. In the next future, it highly advisable to promote organized prostate health testing, which has an important role to play in the community setting whilst ongoing trial and further clinical initiatives continue to find the optimal way to detect and treat this common cancer in the male population.

Author Response

We are very grateful to this reviewer and the other reviewers for their comments.

We accept the need for significant revision of the manuscript to make the central message clearer which we have done in the new version. We have also made a more general discussion rather than commenting on each individual result.

We have also removed some of the data fields that did not reach statistically significant to make the tables clearer as well.

We think that this has resulted in a simpler and clearer paper in keeping with this reviewers central requirement and all the reviewers’ comments.

Reviewer 4 Report

Comments and Suggestions for Authors

The authors have written an interesting article on community-based PSA-testing. I do think that the article can be improved.

·Abstract, objective: would it have been possible to reach out to men who decided not to use the service? And learn more about why this specific group of men doesn’t want to participate in such a community-based PSA-testing initiative? That could be very informative for a potential, future national screening program.

·Background, line 69-70: It is described that the GFCT has grown into a large-scale testing program with over 130.000 men tested and more than 230.000 tests conducted. Could the authors give some context? It sounds like a lot, but not all readers will be familiar with the demographics of the UK, i.e. how many men are eligible to get a PSA-test according to the GFCT versus the 130.000 that do get tested.

·Background, line 75-76: can the authors provide some more information on the ‘Riskman’ algorithm and the ‘Riskman’ score? Please include a reference.

·Methodology, line 87-88: data is not anonymized  if the GFCT can still track back whom completed which survey. Then the data is pseudonymized.

·Methdology, line 88-89: men were invited to complete an online questionnaire. Did men need to sign informed consent? Has an ethics committee looked and approved the study proposal?

·Methodology, lines 87-89: a 13-item questionnaire is mentioned. What items did the questionnaire include? Is the questionnaire validated?

·Methodology, lines 89-90: is anything known about the group of men that did not respond to the questionnaire? Since, if I understand correctly, all men participated in their PSA testing event.

·Methodology, lines 91-93: ‘PSA results were classified into low risk-normal PSA level…’. Are the thresholds used in line with guidelines available of for instance the European Association of Urology and/or national guidelines?

·Methodology, lines 102-104: please include a reference for the IPSS.

·Methodology, lines 104-106: why did the authors collect data on hand patterns, baldness at 40s, information on family history of PCa? What was the rationale behind it?

·Methodology, table 1: where is the follow-up based on?

·Discussion, lines 299-305: I was wondering how it works with the charities and the national healthcare system? How do they relate or work together? If a man visits the GFCT program, how does it work cost wise? How does it work if a prostate cancer is diagnosed through the GFCT program? Will a man be referred for treatment to the NHS?

·Discussion, lines 312-314: if the authors refer to results, of in this case The Riskman algorithm, then please include a reference or references. At multiple places in the article references need to be added to substantiate a statement that is being made.

Overall: after reading the objective of the study i.e. ‘This study aims to explore data gathered from the GFCT public survey as to participants attitudes to the service and to evaluate the approach adopted’, I had different expectations of what would be described in the manuscript. Personally it feels a bit like everything that can be discussed is discussed, without a clear rationale or a clear red tread. 

Author Response

(The authors gave the same response as above.)

Round 2

Reviewer 1 Report

Comments and Suggestions for Authors
  1. The revision of risk group classification is insufficient. The authors need to clarify the basis for the percentage of risk (low risk: 0-49%, intermediate: 50-74%, high: 75-100%) and the PSA range set. In addition, if the upper limit of the PSA threshold is to be judged as "red," then wouldn't the upper limit for high risk in the table be unnecessary?
  2. ROC curve:The value of risk classification cannot be understood unless you show not only the ROC curve for all cases but also the ROC curve for each risk.
  3. Although this study reports the results of social initiatives, I think that the content of the discussion greatly deviates from what can be derived from the results.

Author Response

We would like to thank Reviewer 1 for their valuable comments. We have made the following changes accordingly.

1. The revision of risk group classification is insufficient. The authors need to clarify the basis for the percentage of risk (low risk: 0-49%, intermediate: 50-74%, high: 75-100%) and the PSA range set. In addition, if the upper limit of the PSA threshold is to be judged as "red," then wouldn't the upper limit for high risk in the table be unnecessary?

  • Thank you, we have corrected all these points. We removed the %’s and removed the upper boundary from red group.

2. ROC curve:The value of risk classification cannot be understood unless you show not only the ROC curve for all cases but also the ROC curve for each risk.

  • We have added calibration plot. We did not perform the ROC for each risk group as the model had been developed elsewhere (Lophatananon, 2024) where its development is shown in full rather here we recoded our data to be prostate cancer or non-prostate cancer and input selected variables into multivariable regression model to seek their performance.

3. Although this study reports the results of social initiatives, I think that the content of the discussion greatly deviates from what can be derived from the results.

  • We have added the discussion of our results as suggested.

Reviewer 2 Report

Comments and Suggestions for Authors

Thank you for your revisions. However, some of my previous comments remain unaddressed. You may need to consult a biostatistician regarding the methodological issues.

Below are my specific questions and suggestions. Please provide a point-by-point response.

Abstract

  1. Line 18: Change “The Graham Fulford Charitable Trust (GFCT)’s” to “the GFCT’s”.
  2. Line 20: Change “assesses” to “assess”.
  3. Line 23: Compared to what?
  4. Line 24: It is clear which results support the claim that GFCT’s program had high uptake and satisfaction.
  5. Line 25: No results were provided to support the claim that the Riskman score improves predictive accuracy over PSA alone.
  6. The conclusion should align with your objectives and results.

Introduction

  1. Line 41: A paper on breast cancer screening is not relevant here.
  2. Line 37: Explain the abbreviation “PSA”.
  3. Line 53: Change “prostate-specific antigen (PSA)” to “PSA”.
  4. Line 59: What is “PCa”?
  5. Lines 50-76: This text may still be shortened.
  6. Line 80: Change ““Riskman”” to “Riskman”.
  7. Line 80: Define the abbreviation “PSAft%”.
  8. Line 81: Change “Recent further devlopments” to “Further developments”.
  9. Objective: There are no results provided to evaluate participant attitudes.

Data collection

  1. Indicate the time elapsed between PSA testing and completing the questionnaire.
  2. Do you have an estimate of the time between PSA testing and cancer diagnosis?
  3. Include the questionnaire in an appendix.
  4. Lines 86-87: Reporting the number of invited and participated men belongs in the Results-section.
  5. Line 89: Define “IPSS”.
  6. Lines 97-98: It is not clear, whether men were informed about their cancer risk based on the PSA-age cut points or Riskman score. If Riskman score was already available for these men, how was it calculated?

Data analysis (Consult a biostatistician if necessary)

  1. Describe how a Riskman score is calculated. (Is it a predicted probability obtained using a logistic regression with three predictors? What cutoff for this probability is used to triage men into different risk groups?)
  2. Clarify whether Age, PSA, and PSAft were introduced into the logistic regression model as categorical or continuous variables. If continuous linear: how did you assess linearity between independent variables and log odds outcome?
  3. Did each participant complete a survey only once? If not, explain how the analyses were accounted for repeated measurements.

Results

  1. Report the number and percentage of men invited, responded, and included in the main analysis.
  2. To address selection bias, include a table (in the appendix) comparing characteristics of participants and non-participants.
  3. Line 116: Specify the number of participants.
  4. Table 2: Present characteristics separately for cancer and non-cancer groups.
  5. Table 2: Change “Cancer grade group” to “Prostate cancer grade group”.
  6. Table 2: Use "No" instead of "no".
  7. Table 2: Indicate how many men had missing PSA and PSAft% results.
  8. Table 2: Add "Unknown" category for variables with missing data.
  9. Table 2: Specify the units of measurement for continuous variables.
  10. Line 119: These two sentences are redundant and can be omitted.
  11. Line 124: Add the total number of observations (N=2048).
  12. Table 3: Change “Numbers” to ”No”
  13. Table 3: Change “Std.Dev.” to “SD”
  14. Line 126: Change “Mean PSA” to “PSA statistics”.
  15. Line 126: Add the total number of observations (N=2062).
  16. Why does Table 3 have fewer observations than Table 4? Shouldn’t it be the other way around?
  17. Table 4: Change “Std.Dev.” to “SD”
  18. Tables 2-4: Define “SD” in the legends.
  19. Lines 131, 136: Table 5 doesn’t show the risk estimates for age, PSA and PSAft%, but rather the risk estimates for prostate cancer.
  20. Line 136: Add the total number of observations (N=?).
  21. Line 136: Change “multivariate” to ”multivariable”.
  22. Table 5: Change “95% C.I.” to “95% CI”
  23. Table 5: P-value column can be omitted since CIs provide more information.
  24. Revise the Figure 1’s title for clarity: "Receiver Operating Characteristics (ROC) Curve for prostate cancer model including age, PSA, and PSAft%."
  25. Add a y-axis label to Figure 1.
  26. Provide CIs for the AUC.

Discussion

  1. Begin the Discussion with the main findings of this study, aligning them with the study’s objectives and results.
  2. As Discussion lacks clear structure, consider organizing it into the following sections: Main findings, Strengths and limitations, Comparison with other studies, Implications, Conclusions.
  3. Address selection bias.
  4. Discuss the response rate and the extent of missing data.
  5. Assess potential information bias due to the time lag between testing and completing the questionnaire.
  6. Discuss the limitations of the methodology used to evaluate the prediction model and the need for further validation [ref: Shipe, M., et. al., Developing prediction models for clinical use using logistic regression: an overview. J of Thoracic Disease, 11(4), 2019]
  7. Lines 160, 201, 204: Statements regarding cost-effectiveness must be supported by cost-effectiveness analyses.
  8. Line 160: Delete “(free-to-total PSA ratio)”.
  9. Line 161: No results were provided to support the claim that the Riskman score improves predictive accuracy over PSA alone.
  10. Line 250: Change “albeit it” to “albeit in”.

Conclusions:

  1. Conclusions should reflect the study’s objectives and results.

Reviewer 4 Report

Comments and Suggestions for Authors

The authors are thanked for incorporating feedback from the reviewers. However, I still have some questions left. 

  1. In table 2 the general characteristics of the respondents are shown. For age a min-max age of 34-90 is reported. Why would you measure PSA for a man aged 34? The same question applies for a man aged 90 years.
  2. The authors describe that they have used the Riskman Score. Playing advocate of the devil, why would you develop a 'new' risk assessment instrument? Why not use one that is publicly available and that you maybe only have to calibrate to the target population? 
  3. Why is the charity using different PSA cut-offs for different ages? Please also have a look at the risk-algorithm set-up and used within the PRAISE-U consortium. 
  4. To me it is still not clear how big a part of the make population the charity is providing PSA-tests to. 
  5. The authors mention the TRANSFORM-trial that will start in the UK. Will there be overlap between men that would like to participate via charity and those who will receive an invitation via TRANSFORM? 
  6. The references in the text are in the format of a last name and year. While in the reference list numbers are used. Please use the method that is advised by the journal. 

Author Response

The authors are thanked for incorporating feedback from the reviewers. However, I still have some questions left. 

-We are grateful for the suggestions.  Thank you.

  1. In table 2 the general characteristics of the respondents are shown. For age a min-max age of 34-90 is reported. Why would you measure PSA for a man aged 34? The same question applies for a man aged 90 years.

- This is now Table 3, and another reviewer suggested reformatting its presentation. Regarding the age of survey respondents, GFCT did not exclude men from PSA testing based on age; therefore, the age range includes both younger and older individuals.

  1. The authors describe that they have used the Riskman Score. Playing advocate of the devil, why would you develop a 'new' risk assessment instrument? Why not use one that is publicly available and that you maybe only have to calibrate to the target population? 

- We previously published the TIERED approach (Lophatananon, 2024), and this has been incorporated into the discussion. While we agree with the reviewer's comments, our aim was to explore an existing inexpensive method for risk assessment and evaluate its potential implementation in charity-led initiatives such as those conducted by GFCT.

  1. Why is the charity using different PSA cut-offs for different ages? Please also have a look at the risk-algorithm set-up and used within the PRAISE-U consortium. 

- The charity has revised its cut-off points based on scientific evidence. The cut-off points presented in this paper reflect those used at the time of data collection, but they have since been updated.

  1. To me it is still not clear how big a part of the make population the charity is providing PSA-tests to. 

-This is now shown in Table 1 and Figure 1 to better show the reach of the Charity.

  1. The authors mention the TRANSFORM-trial that will start in the UK. Will there be overlap between men that would like to participate via charity and those who will receive an invitation via TRANSFORM? 

- There is potential for participant contamination in the trial.  GFCT however has been involved in the discussions of this with TRANSFORM trial team and recommended that men who had a PSA test with them should inform the trial and not be recruited and that different regions of the county where GFCT testing rates are lower be used.

  1. The references in the text are in the format of a last name and year. While in the reference list numbers are used. Please use the method that is advised by the journal. 

- We have corrected this, thank you.

Round 3

Reviewer 2 Report

Comments and Suggestions for Authors

attached

Reviewer 4 Report

Comments and Suggestions for Authors

After several rounds of revisions, the manuscript has improved. Please take one more look at statements that inquire a reference. 

At several places in the manuscript, the authors should add reference. For instance when they mention that the IPSS is used, a reference to the IPSS should be included. Second example: 'The Riskman algorithm was initially developed outside this study and documented in a separate publication by the lead author.' --> refer to the publication then.

Author Response

After several rounds of revisions, the manuscript has improved. Please take one more look at statements that inquire a reference. 

At several places in the manuscript, the authors should add reference. For instance when they mention that the IPSS is used, a reference to the IPSS should be included. Second example: 'The Riskman algorithm was initially developed outside this study and documented in a separate publication by the lead author.' --> refer to the publication then.

-Thank you for reviewer 4 comments, we have checked and added references as requested.